# Are Ecosystem Services Provided by Street Trees at Parcel Level Worthy of Attention? A Case Study of a Campus in Zhenjiang, China

**DOI:** 10.3390/ijerph20010880

**Published:** 2023-01-03

**Authors:** Guangxi Shen, Zipeng Song, Jiacong Xu, Lishuang Zou, Lijin Huang, Yingnan Li

**Affiliations:** 1Department of Environmental Design, School of Arts, Jiangsu University, Zhenjiang 212013, China; 2Department of Environmental Horticulture, University of Seoul, Seoul 02504, Republic of Korea; 3School of Art and Design, Dalian University of Technology, Dalian 116034, China; 4Department of Environmental Planning Institute, Seoul National University, Seoul 08826, Republic of Korea

**Keywords:** ecosystem services, climate regulation, urban forests, street tree, parcel level, i-Tree Eco, sustainable development, economic value

## Abstract

Street trees in urban areas have positive impacts on the environment, such as climate regulation, air purification, and runoff mitigation. However, the ecosystem services (ES) provided by street trees at the parcel level remain a notable gap in the existing literature. This study quantified the ES benefits provided by street trees at Jiangsu University in Zhenjiang, China, which could represent the parcel level. A widely applied model, i-Tree Eco, was used to evaluate the ES, including carbon storage, carbon sequestration, pollutant removal, rainwater runoff reduction, and their economic value. We also analyzed how these are affected by the structure of tree species. The results revealed that the 10 most abundant species accounted for 84.3% of the total number of trees, with an unstable structure of species composition. A reasonable age proportion was important since a lower proportion of young trees may make future benefit outputs unstable. The annual economic value provided by ES was USD 205,253.20, with an average of USD 79.90 per tree. *Platanus orientalis* had the highest single plant benefit, indicating that the management pattern of street trees could be adapted in different environments to maximize benefits. Thus, street trees at the parcel level should receive more attention with strategic planning and management in order to maximize the ES and their economic value.

## 1. Introduction

Urban green spaces play an important role in alleviating urban ecological problems [1]. Trees convert carbon dioxide (CO_2_) into oxygen (O_2_) through photosynthesis and by filtering pollutants such as nitrogen dioxide (NO_2_), sulfur dioxide (SO_2_), ozone (O_3_), carbon monoxide (CO), and particulate matter less than 10 µm (PM_10_) and less than 2.5 µm (PM_2.5_) [2,3], which are the main factors contributing to climate change. Trees can mitigate the impact of the heat island effect by cooling the air through shading and transpiration [4], which is particularly important in the context of global warming. Urban expansion and large-scale land acquisition for urban development have changed natural, water-permeable surfaces into impervious ones [5]. The widespread presence of impermeable areas has reduced the rainfall infiltration capacity [6], resulting in extensive flooding in urban areas. Tree canopies intercept rainfall, which infiltrates the soil and reduces surface runoff [7]. 

Many street trees in cities were originally planted to beautify urban areas. However, with ongoing urbanization, the ecological [8,9], social [10,11], and economic [12] impacts of street trees are receiving increasing attention. Research on the ecosystem services (ES) they provide and the economic value they generate have become key topics in developed regions, such as the United States [13,14,15], Europe [16], and Japan [17]. Due to the rapid development of some countries, many cities are facing considerable growth in urban construction [18] and urban roads [5]. The strategic planning of street tree planting to improve the urban ecological environment has also been a key focus in developing countries [19,20,21]. Wang et al. [22] studied the structure and ES provided by street trees in Dalian, China, and quantified the economic value generated. They found that urban street trees are a fragile resource, but the economic value they provide is higher than the management costs. Chen et al. [23] investigated the differences in the carbon value of graded street trees along an urban–suburban gradient in Shanghai, China. They found pronounced differences in the carbon reduction contributions between urban and suburban street trees and recommended that special attention be paid to the local degree of urbanization in future construction projects. However, Richardson and Shackleton [24] reported that street trees in South Africa are vulnerable to damage, suggesting that some regions face unique challenges in implementing street tree planting and maintenance strategies to optimize ecosystem services and reduce the cost of planting. Given that street trees are located in highly developed urban areas, they are often subject to biophysical and anthropogenic stresses and have higher mortality rates than the same species in non-urban areas [25]. Therefore, it is essential to investigate how to optimize the composition and structure of street tree planting, thereby improving street tree health and further enhancing the provision of ES. 

In addition to the challenges associated with the survival of street trees in urban areas, trees can also have environmental impacts in certain situations, for example urban trees can mitigate the effects of climate change by sequestering carbon (removing carbon dioxide from the atmosphere) and reducing energy use in buildings [26]. However, when trees die, the stored carbon is also released back into the air, thus negatively impacting the climate. Trees also release volatile organic compounds (VOCs), which are precursor chemicals that form ozone and other pollutants [27]. VOC emissions are harmful to human health and the climate, and species selection for new construction projects should take this into account. VOC emissions depend not only on the species but also the leaf biomass and local meteorological conditions within a given species [28].Therefore, in addition to species selection, proper management strategies should be implemented to minimize VOC emissions in new and existing street tree groves and urban forests. 

The ES provided by street trees at the urban scale or by some typical types of urban green areas are well recognized, such as urban forests [29], residential areas [30], government institutions [31], and parks [32,33]. These studies also demonstrated the economic value of ES, thereby identify the strategies on how to plan the inclusion and select appropriate species in urban green areas. However, there is limited research that has focused on the ES and the value of street trees at the parcel level. University campuses are important components of green infrastructure that contain large numbers of street trees, which is also a typical type of land that represents the parcel level. Investigating the ES of street trees at university campuses not only provide us an insight on how much benefit they can provide at a parcel level but can also be a reference for greening mangers on tree species selection and tree management. 

This study aimed to assess and quantify the structural, air quality, and climate-related ES of street trees at Jiangsu University (JSU). The analysis was carried out using the i-Tree Eco model, which was applied for the first time in the study of ES in Zhenjiang, Jiangsu, China. A campus-based study was conducted to investigate the value and structure of street trees at the parcel level.

Specifically, this study aimed to answer the following two questions: 

(1) How many ES can campus street trees at the parcel level provide, and do parcel-level ES differ from urban-scale ES enough to warrant further investigation?

(2) How does the structure of campus street trees at the parcel level affect ES, and what are the best strategies for street tree planting and maintenance at the parcel level?

## 2. Materials and Methods

### 2.1. Study Area

This study was conducted at JSU, located in Zhenjiang, Jiangsu Province, China (32°11′ N, 119°30′ E). The total campus area studied was approximately 440 acres, with a vegetation cover of 61.82 ± 3.28%, as measured by the i-Tree Canopy version 6.1 (https://canopy.itreetools.org/ ((accessed on 12 July 2022)). We used ArcGIS 10.2 (https://www.arcgis.com/ (accessed on 25 July 2022)) to obtain road data from Google Maps, which showed that the total length of the roads on campus was 28,390.2 m. All street trees on campus roads were sampled for the analysis. The staff residential area on the east side of the campus (marker: blue block slash) was not included in this study because it was subjected to different management protocols (Figure 1).

Zhenjiang has a north subtropical monsoon climate with a distinct monsoonal nature and four distinct seasons. The average annual temperature is 15.6 °C, the average monthly maximum temperature is 27.4 °C in July, and the minimum temperature is 2.7 °C in January. The average annual precipitation is 1500 mm, with the most precipitation received in July (average of 179 mm), and the least precipitation received in December (average of 24 mm) (National Meteorological Information Centre—China Meteorological Data Network (CMA.cn)).

### 2.2. Ecosystem Service Value: i-Tree Eco Model Application

The i-Tree Eco model was selected to evaluate the ES benefits of street trees in this study. It is a state-of-the-art, peer-reviewed software suite from the United States Department of Agriculture Forest Service that delivers urban and community forestry analysis and benefits assessment tools (www.itreetools.org). The i-Tree tools helps communities of all sizes strengthen their urban forest management and outreach efforts by quantifying the environmental services provided by trees and assessing the structure of urban forests [34]. A considerable number of studies have proven its reliability and soundness [17,35,36,37,38,39]. Simulations can be carried out in any area with the data already available. The collected street tree data were combined with local environmental data and used to estimate the ES provided by the street trees. The outputs from i-Tree used in this study included carbon storage and sequestration, air pollutant removal and emission of VOCs, and stormwater runoff reduction. In addition, the value of annual ES benefits was also calculated based on the default economic data from the i-tree eco system [40]. The relevant specific prices and valuation methods are reflected in Section 3.2.

#### 2.2.1. Tree Data Collection

A comprehensive census of all the street trees in the 440 acres of JSU was conducted between April and June 2022, with all the field surveys conducted according to the i-Tree Eco v6 guidelines [41]. All trees with a diameter at breast height (DBH) greater than or equal to 4.5 cm and a height greater than 1.3 m were included in this survey. The instruments used for the survey included a distance measuring device, calipers, and diameter tape. DBH was measured using calipers or diameter tape at 1.30 m above ground level. Trees were identified at the species level in the field with the help of regional botanical monographs [26]. Specimens and photographs were collected for plants that were unidentifiable in the field for later consultation with a specialist botanist for identification. Data collected included (1) species, (2) diameter at breast height, (3) tree height, (4) crown size, (5) crown loss, (6) die-back rate, and (7) crown exposure. These data were recorded in Microsoft Excel 2019 to assess the street tree structure and associated ES and values.

#### 2.2.2. Hourly Weather Data and Pollution Data

Due to an increase in the number of international users of the i-Tree suite, new users can now uploading their data on study sites to quantify ES, analyze urban forest structures, and include local hourly air pollution and meteorological data [42]. However, the model’s original repository does not currently contain data for any cities in Jiangsu Province, China. In this study, we introduced the city of Zhenjiang, Jiangsu Province, into the i-Tree database, including location information, precipitation data, and pollution data. Hourly pollutant data for 2019, including CO, NO_2_, O_3_, PM_10_, PM_2.5_, and SO_2_, were provided by the Zhenjiang Vocational Education Center monitoring station (119.491 N, 32.215 E). Hourly rainfall data were obtained from the Nanjing Lukou meteorological station (31.742 N, 118.862 E), which is located 75 km away with similar precipitation to our study site.

## 3. Structure and Function

### 3.1. Street Tree Structure

#### 3.1.1. Importance Value

The mean values of three indicators (percent total number of trees, percent total leaf area, percent canopy cover) were used to calculate importance values (IV) for each species. This value indicated the dependence of the study area on a particular species, ranging from 0–100, with 100 representing complete dependence on the species [43]. 

#### 3.1.2. Age Structure

In our study, street trees were classified by DBH into four categories, namely young trees (0–15 cm), mature trees (15–30 cm), mature trees (30–60 cm), and old trees (>60 cm) [44], so as to determine the reasonableness of their age structure.

### 3.2. Estimation of Ecosystem Services by Street Trees

#### 3.2.1. Carbon Storage and Sequestration

Carbon storage was derived from the measured tree data with the biomass equations obtained from the literature, although since our study environment was urban, the biomass values were reduced by 20%. To estimate the gross amount of carbon sequestered annually, the average diameter growth from the appropriate genera, diameter class, and tree condition were added to the existing tree diameter data (year x) to estimate the tree diameter and carbon storage in year x + 1. For this analysis, the carbon storage and carbon sequestration values were calculated for USD 178 per metric ton [41].

#### 3.2.2. Air Pollutant Removal

A description of the trees, including percentage cover, leaf area index, and percentage evergreen, were combined with field-measured air pollutant and meteorological data to estimate the number of pollutants removed, including CO, NO_2_, O_3_, PM_2.5_, PM_10_, and SO_2_ [45,46].

The emission of VOCs also has an impact on the climate, the most direct effect being the production of ozone. The amount of VOC emissions depends on the vegetation genus, leaf dry-weight biomass, air and leaf temperature, and other environmental factors. VOC emissions were calculated with adjustments for leaf biomass and pollutant emissions [47].

For this analysis, the pollution removal value was calculated based on the prices of USD 1508 per metric ton (CO), USD 1061 per metric ton (O_2_), USD 10,619 per metric ton (NO_2_), USD 2599 per metric ton (SO_2_), USD 7090 per metric ton (PM_2.5_), and USD 7090 per metric ton (PM_10_).

#### 3.2.3. Runoff Reduction

The annual runoff mitigated by street trees was calculated based on the precipitation intercepted by vegetation and the estimated difference between annual runoff with and without vegetation [48]. Leaves, branches, and bark can intercept precipitation and mitigate surface runoff. However, for this analysis, only the precipitation intercepted by leaves was considered. To estimate the impact of trees on surface runoff, estimates were made based on the current situation with and without trees. For this analysis, the value of runoff avoided was calculated based on the price of USD 2.30 per m^3^.

## 4. Results

### 4.1. Structure of Street Trees at JSU

#### 4.1.1. Species Composition

A total of 2569 street trees at JSU were identified, comprising 36 species in 27 genera and 24 families. Deciduous trees were the dominant leaf phenology (75%), while 25% were evergreen. The 10 most abundant species accounted for 84.3% of trees sampled (Table 1). The six most abundant species (Figure 2) were *Cinnamomum camphora* (43.1%), *Koelreuteria paniculata* (13.2%), *Prunus subhirtella* (4.9%), *Ginkgo biloba* (4.7%), *Platanus orientalis* (4.7%), and *Styphnolobium japonicum* (4.6%).

#### 4.1.2. Importance Values

The combined IV for the 10 most abundant tree species was 89.5 in this study. *C. camphora* had the highest values for percentage total tree numbers, percentage total leaf area, and percentage total canopy cover. Therefore, it had the highest IV of 43.3, which showed a high degree of dependence. *K. paniculata* and *P. orientalis* followed, with IVs of 18.6 and 10.1, respectively. Rosaceae, the most commonly identified family, had a total IV of 2.4 (Table 1).

Leaf area substantially influenced the IV of the investigated species. The six most abundant species ranked by total leaf area were *C. camphora* (40.2%), *K. paniculata* (21.1%), *P. orientalis* (17.4%), *P. stenoptera* (5%), *G. biloba* (2.1%), and *S. japonicum* (2.1%) (Table 1). *P. orientalis* had a maximum average crown area and leaf area of 90 m² per tree and 1257 m² per tree, respectively. The average leaf area of *P. orientalis* was followed closely by *P. stenoptera* (973 m²), *C. deodara* (665 m²), and *K. paniculata* (540 m²). The most abundant tree species was *C. camphora*, with an average leaf area of 315 m² (Figure 3).

#### 4.1.3. Age Structure

The age structure of the campus street trees was relatively uneven, with young trees (0–15 cm DBH) accounting for 19.1%, maturing trees (15–30 cm DBH) accounting for 57%, and mature trees (30–60 cm DBH) accounting for 19.5%. The proportion of old trees (>60 cm DBH) was the smallest at 4.4% (Figure 4). Among the 10 most abundant species, 21.3% were young trees, 53.0% were maturing trees, 13.2% were mature trees, and 12.6% were old trees. The proportion of old trees in this sampling was relatively close to the ideal age structure. The largest proportion of young trees of *P. subhirtella* was 85.7%, but there were no trees with a DBH of more than 30 cm (Figure 4). The percentage of young trees and old *K. paniculata* trees was zero. The proportion of old *P. orientalis* trees in was 54.2%, followed by *P. stenoptera* at 77.8%. The number of old trees of other species was relatively low.

### 4.2. Ecosystem Services Provided by Street Trees at Jiangsu University

#### 4.2.1. Carbon Storage and Sequestration

The sampled street trees were estimated to store 657,700 kg (USD 122,438) of carbon in their biomass. Among the most abundant tree species, *C. camphora* (49.9%) stored the most carbon, followed by *K. paniculata* (18.7%), *P. orientalis* (15.4%), and *P. stenoptera* (4.4%). The four most abundant species all exceeded the average carbon storage per tree. The gross sequestration per year was approximately 59,490 kg (USD 75,300). *C. camphora*, *K. paniculata, P. orientalis*, and *S. japonicum* sequestered the highest percentage of carbon annually, approximately 56.9%, 12.1%, 7.8%, and 3.4%, respectively. The average amount of carbon stored and sequestered per tree was 256 kg and 23.2 kg, respectively. The economic value provided by carbon storage and sequestration was USD 77.00/tree on average. The species that produced the highest net value were *P. orientalis* (USD 205.70/tree) and *P. stenoptera* (USD 161.40/tree). However, *G. biloba* (USD 12.90/tree), *P. subhirtella* (USD 27.00/tree), and *M. grandiflora* (USD 25.00/tree) were substantially lower than average (Table 2).

#### 4.2.2. Air Pollutant Removal and Emission of VOCs

The campus street trees removed approximately 2120 kg of air pollutants in a full year, with an associated value of USD 18,038.80 (Table 3). Pollution removal was the greatest for O^3^, accounting for 43.6% and 53.4% of the total removal and economic value, respectively (Figure 5).

We found that *C. camphora* had the highest pollutant removal value of USD 7807.20/y (43.3%), followed by *K. paniculata* (USD 4096.80/y, 22.7%) (Table 3). These two species also removed the most pollutants at 850 kg/y (40%) and 450 kg/y (21.2%), respectively. The trees on campus streets released an estimated 255.9 kg/y of VOCs (Figure 6). Among the most abundant tree species, *P. orientalis* emitted the highest quantity of VOCs at 124.4 kg/y (48.6%). The largest contributor to air pollution removal was *C. camphora*, and it emitted 3.1 kg/y (1.2%) of VOCs. The second largest contributor to air pollution removal, *K. paniculata*, did not emit any measurable VOCs. On a per-tree basis, the quantity and value of air pollutant removal were 0.82 kg/y and USD 7.02/y, respectively, while VOCs emissions were 0.12 kg/y. Among the dominant tree species, *P. orientalis* (3.1 kg/y, USD 28.10/y) provided the most value and was the most effective at pollutant removal, but it also emitted the most VOCs (1.04 kg/y).

#### 4.2.3. Stormwater Runoff Reduction

The sampled street trees intercepted approximately 2352.7 m^3^ of rainfall annually, and this reduction in stormwater runoff resulted in an economic value of USD 5582.60. Among the 10 most abundant species, *K. paniculata* accounted for 21.3% of the total stormwater runoff reduction, followed by *P. orientalis* with 17.6%. The average value was USD 2.17/per tree. Three of the ten most abundant species, *K. paniculata* (USD 3.48/tree), *P. orientalis* (USD 8.09/tree), and *P. stenoptera* (USD 6.26/tree), exceeded the average (Table 4).

### 4.3. Overall Performance of Street Trees on Ecosystem Services 

The total value of the street trees sampled was evaluated by summing the values of the three different estimated ES benefits, which were calculated at USD 164,415.20 annually, or USD 64.07 per tree on average. Excluding carbon storage, pollution removal contributed the most, accounting for 53.7% of the remaining economic value. Reducing runoff contributed the least at 15.3% (Table 5). Among the 10 most abundant species, *C. camphora* had the highest total value of USD 784,447.00, followed by *K. paniculata* (USD 29,958.60), *P. orientalis* (USD 24,316.50), and *P. stenoptera* (USD 7071.40). On a per-tree basis, *P. orientalis* (USD 202.60), *P. stenoptera* (USD 157.10), *K. paniculata* (USD 88.40), and *C. camphora* (USD 70.80) provided the highest value (Figure 7).

## 5. Discussion

### 5.1. Analysis and Recommendations for Strategic Planting of Street Trees

The results showed that among the 2569 trees collected, there were 24 families, 27 genera, and 36 species. This result was comparable with other studies on campus street trees. For example, more than 21 families, 28 genera, and 31 species were measured at Northwest Agriculture and Forestry University (NAAFU) in China [49]. The ES benefits of NAAFU were much lower than those reported in similar studies, which may be attributed to the lower species diversity [50]. Two street tree surveys at Nanjing Forestry University (NJFU) independently identified fewer than thirty-one families, forty-nine genera, and fifty-four species [51]. *Platanus acerifolia* had the highest per-plant benefit in the NJFU study, while *P. orientalis*, also in the Pendulaceae family, demonstrated high ES benefits in our study. The most abundant species, which strongly overlapped with our study, included *C. camphora, K. paniculata*, *G. biloba*, and *P. persica.* This may explain why the final total benefit values from the two studies were relatively similar. In this study, the 10 most abundant species accounted for 84.3% of all trees sampled, with *C. camphora* accounting for 43.1% and Rosaceae accounting for 25% of the total number of families. This was inconsistent with the generally accepted rule of diversity, with no single species accounting for more than 10% of the population, no genus for more than 20%, and no family for more than 30% [43]. Similar contradictions have been found in many urban areas [20,52], parks [53], and campuses [54]. The diversity of street trees plays an important role in enhancing the resilience of street ecosystems and reducing environmental damage from natural hazards [22,55,56]. They can also have aesthetic and psychological benefits [57]. Griess et al. [58] found that plants have a greater chance of survival when grown in diverse stands. Therefore, it is necessary to increase street tree species diversity, and diversity targets cannot only be based on the proportions of species families and genera in a given site type [59]. Finding solutions to this problem requires a strong focus on stratigraphic-level street tree studies.

Our study found that *C. camphora* and *K. paniculata* had the highest leaf area and canopy cover, with IVs of 43.3% and 18.6%, respectively. However, overdependence on *C. camphora* is problematic in China’s coastal areas [60], where damage by invasive pests of camphor trees is relatively common [61,62]. Due to the humid climate and relatively high temperatures in the eastern part of the country near the coast, Luo and Lau [63] suggested that campus streets in such climatic zones should reduce the planting of this species to avoid disproportionately large IVs. We also found that *P. orientalis* had a much higher IV than species with similar abundance. This was because it had the highest average leaf area at 1257.3 m², which was higher than in species in the Southern Caucasus Region [64] and in street trees in Kyoto, Japan [17]. This may be because trees in urban streets are frequently pruned, reducing their average leaf area. As managers in different locations may have different pruning strategies, the same tree species may have varying average leaf areas and consequently varying IVs. The complex urban environment, with its presence of obstacles, size of tree pits, and use of adjacent land, is also an important influencing factor [25]. Therefore, appropriate adjustments are most effectively made to the management pattern of street trees at the parcel level.

Young and old trees accounted for only 19.1% and 4.4% of our total sample, respectively, though a few abundant species had high proportions of old trees, namely *K. paniculata* (100%), *P. stenoptera* (77.8%), and *P. orientalis* (54.2%). A low proportion of old trees may lead to poor ES. Large trees can provide considerably more ES than small trees, as Nowak et al. [65] found that healthy old trees sequestered 530 times more carbon than young trees. Similarly, Chen et al. [23] found that the DBH and the canopy width of street trees in Shanghai were the two factors with the most influence on low carbon contribution, with correlation coefficients calculated using i-Tree Eco of 0.72 and 0.68, respectively. The trees in our study performed very well in providing all benefits, which mitigated the impact of having a low proportion of old trees. The high proportion of maturing trees (57%) will ensure that the benefits associated with street trees at JSU will remain relatively stable over longer timescales. However, the low proportion of young trees (19.1%) may not be able to compensate for the loss of benefits from tree loss due to the death of old trees or road widening, which presents a risk in the long term. Therefore, young trees should be planted more frequently and the high mortality rate of young trees should be taken into account [66] to keep the age structure close to an ideal state [67] and to maintain a stable performance.

### 5.2. Ecosystem Service Comparison

Atmospheric CO^2^ reduction is an important aspect in humanity’s response to climate change [68]. Carbon storage and sequestration accounted for 84.7% of the total benefits quantified in our study. The street trees at JSU stored approximately 657,000 kg of carbon, sequestering 59,490 kg of CO^2^ per year. Individual trees stored, on average, 256 kg of carbon and sequestered 23.2 kg of CO^2^ per year. Due to the differences in sample size between studies, we also included the numbers per tree for comparison purposes. Our findings were higher than those at a renovated park in Bangkok, Thailand [53]. The role of the park is predominantly flood control, with the avoidance of runoff contributing over 60% of the estimated ES. Our findings were also higher than surveys on other campuses, for example Shenyang Institute of Technology Campus, Shenyang, China [54], Amity University Campus Noida, Uttar Pradesh, India [69], Auburn University campus, Auburn, Alabama, and North Maharashtra University Campus, Jalgaon, Maharashtra, India [70]. This may be due to a lack of maintenance of trees on campus other than street trees, the greater vulnerability of trees in academic and residential areas to vandalism, and the planting of large numbers of ornamental species for aesthetic reasons. In comparison with studies on street trees at the urban level, the amount of sequestration in our study was lower than in California, USA (34.7 kg/y) [40], and higher than that in Kyoto, Japan (12.5 kg/y) [17].

Trees act as natural biological filters, removing particles from the air and improving air quality. Urban trees can effectively dilute O^3^ in the air. Yu et al. [71] once again confirms that high ambient temperatures strongly amplify the adverse effects of ozone on human mortality. In our study, O^3^ accounted for 43.6% of the pollutant values removed, demonstrating that campus street trees effectively regulated air quality and mitigated the effects of the urban heat island effect. The main source of PM_2.5_ is transport emissions [72], and it has been demonstrated that high PM_2.5_ levels pose a great health risk to urban residents [73]. The percentage of PM_2.5_ removed in this study (4.7%) was not sustainable. This can be explained by Yang et al. [74], which ranked the suitability of common urban tree species for controlling PM_2.5_ pollution. The three most abundant species in our study were all ranked below 40 on this list. From the perspective of protecting university students and employees, the removal of PM_2.5_ from urban forests can be enhanced by selecting more coniferous and broad-leaved species with a high removal efficiency [74]. *P. orientalis* (3.1 kg/y, USD 28.10/y) had the best economic efficiency and pollutant removal. However, it also produced more VOCs, which contributes to the rapid formation of O^3^ [42], creating a potential hazard. Our study showed that the choice of street trees at the stratigraphic level could substantially influence the ecological outcome.

The street trees at JSU could intercept 2352.7 m^3^ of rainfall annually. This is an average interception of 0.9 m^3^ per tree, which was lower than California, USA (2.9 m^3^/tr) [40] and communities in metropolitan Cincinnati, Ohio, USA (6.1 m^3^/tr) [75]. The economic value of the interception was USD 2.17/tree per year, which was the same as Oak Ridge National Laboratory ($2.17/tree) [76]. The total share of value from stormwater interception was 3.4%, which was substantially lower than the results for similar study sites [54]. Due to street trees growing on roads and pavements, the sealed surfaces surrounding the trees increase overall surface runoff [77]. Therefore, it is necessary to adjust the planting plan to increase the number of species with good stormwater interception values. Trees with a larger and denser canopy cover are a suitable choice.

### 5.3. Limitations and Perspectives

In this study, the aesthetic and energy-saving benefits of street trees were not taken into consideration, which may have contributed to bias in the results. Given that i-Tree Streets was replaced by i-Tree Eco, the original aesthetic contribution could not be evaluated, and an alternative approach needs to be further investigated. When considering the selection of street tree species on campus, the ES results should be analyzed in conjunction with students’ preferences. University students’ preferences for campus landscapes vary substantially, with different landscape types producing different behavioral and psychological differences. This study only quantified the ES of street trees on one campus and did not consider differences in climate, geographical location, or the type of campus. These aspects should all be considered in future studies.

## 6. Conclusions

This study assessed the structure and ES of street trees at the parcel level, with the aim of demonstrating how they function as green infrastructure, providing ES and value to society. Our study used universities as an example, and the key findings were as follows:

1. The 10 most abundant species accounted for 84.3% of the total sample size and were overly dependent on a few species, which may lead to a lack of adequate buffers for street trees at the parcel level in the face of catastrophic losses such as pests and extreme weather. The number of camphor trees at JSU should be reduced and managed at an appropriate level. At the same time, species diversity and structural stability should be increased.

2. The outstanding ecological service results of *P. orientalis* showed that the management pattern of street trees at the parcel level could be distinguished from that of urban roads. The implementation of appropriate tree care management practices can maximize the potential of street trees at the parcel level and contribute to the enhancement of the climate.

3. Maturing and matured trees accounted for 76.5% of all the trees in the study, and this age structure could provide more benefits at this stage. However, to ensure a steady output of benefits in the future and to reduce maintenance costs, the proportion of young trees should be increased, and the large old trees should be effectively maintained to extend their lives.

4. The simulation by i-Tree Eco demonstrated that the campus street trees provided a range of ES. The annual value for all street trees was USD 205,253.20 (USD 79.90/tree). Carbon storage and sequestration provide the most economic value at a total of USD 197,738 (USD 77.00/tree). Runoff reduction provided the least economic value at a total of USD 5582.60 (USD 2.17/tree). Among the 10 most abundant species, the largest contributors were *P. orientalis* (USD 202.60/tree), *P. stenoptera* (USD 157.10/tree), *K. paniculata* (USD 88.40/tree), and *C. camphora* (USD 70.80/tree). 

This study combined field surveys and i-Tree Eco simulations to systematically analyze the ecological services and composition of street trees at JSU. Our results indicated that street trees at the parcel level require more attention, and managers can maximize the ecological services provided by street trees based on the information we have provided. This can be beneficial to human health and urban climates. Our research will improve the knowledge about the value of ecological services at the urban parcel level.

## Figures and Tables

**Figure 1 ijerph-20-00880-f001:**
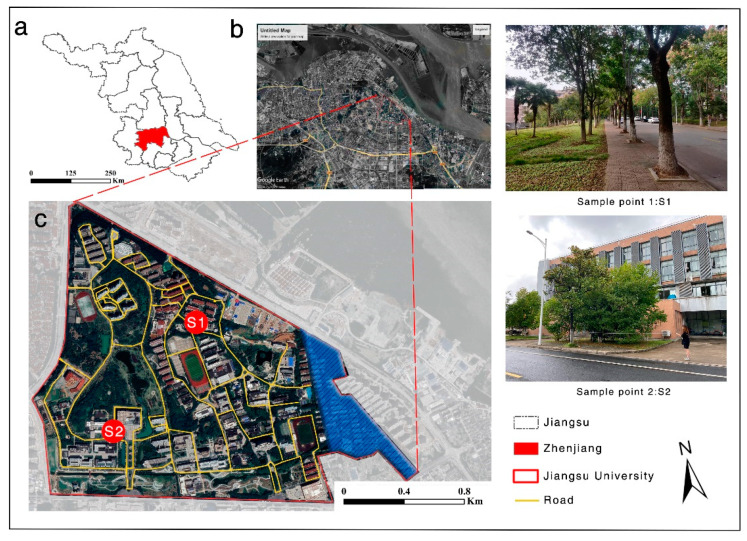
Location of the study site (**a**) Zhengjiang in Jiangsu province, (**b**) JSU in Zhenjiang, (**c**) the region of study area in JSU. The blue block slash indicates the areas excluded from the study, and sampling points are indicated by S1 and S2.

**Figure 2 ijerph-20-00880-f002:**
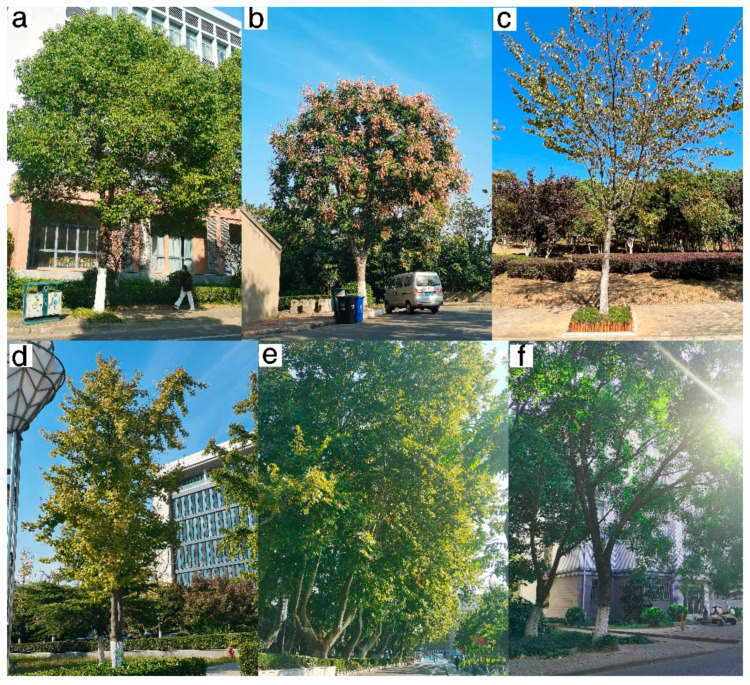
Images of some typical specimens of the six most abundant species street trees at JSU (**a**–**f**). (**a**) *Cinnamomum camphora* (Lauraceae); (**b**) *Koelreuteria paniculate* (Sapindaceae); (**c**) *Prunus subhirtella* (Rosaceae); (**d**) *Ginkgo biloba* (Ginkgoaceae); (**e**) *Platanus orientalis* (Platanaceae); (**f**) *Styphnolobium japonicum* (Leguminosae).

**Figure 3 ijerph-20-00880-f003:**
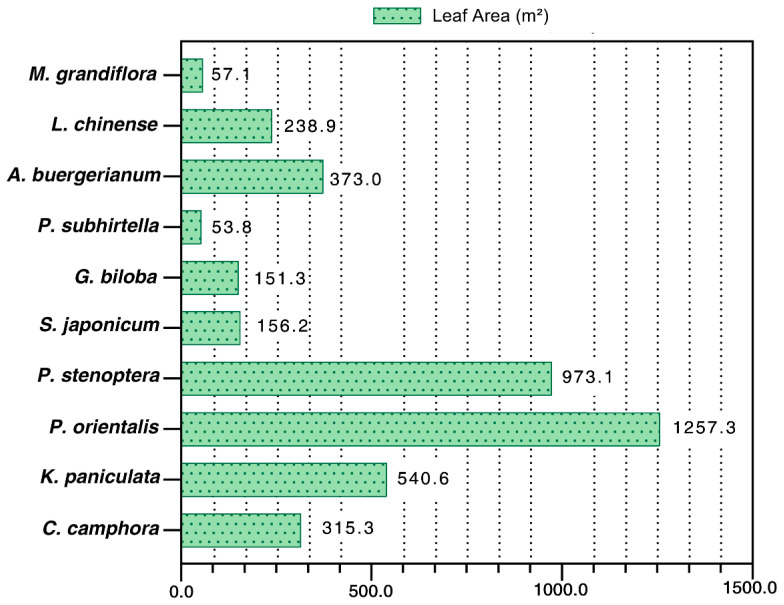
Average leaf area per tree (m²) of the most abundant campus street trees at JSU.

**Figure 4 ijerph-20-00880-f004:**
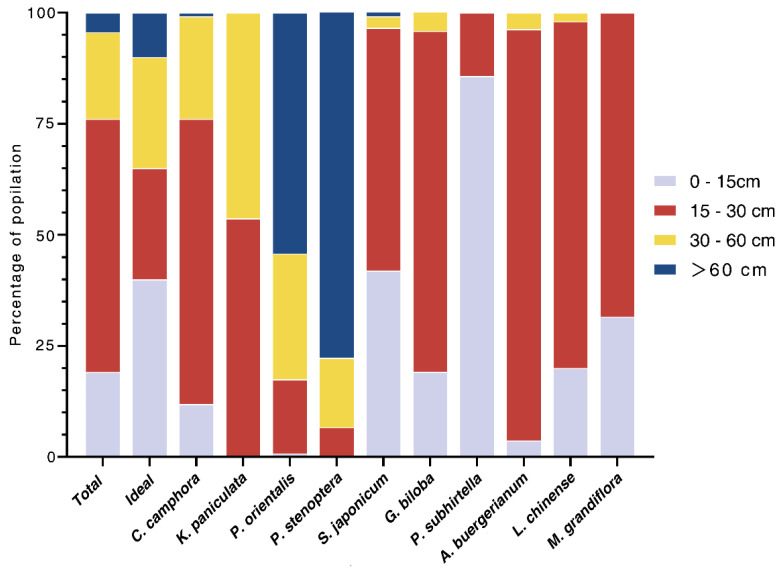
Comparison of the diameter at breast height size distribution of the ten most abundant street tree species against the “ideal” distribution.

**Figure 5 ijerph-20-00880-f005:**
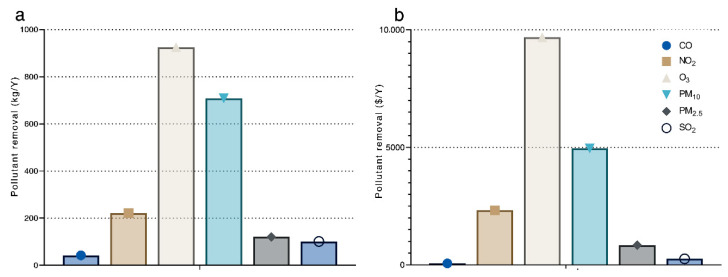
(**a**) Annual pollutant removal and (**b**) annual pollutant removal value.

**Figure 6 ijerph-20-00880-f006:**
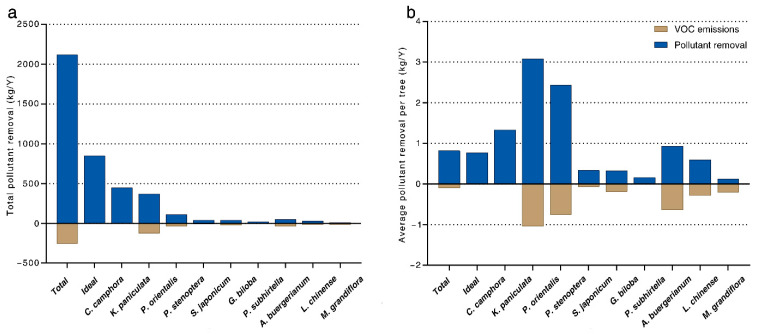
Comparison of pollutant removal and volatile organic compound emissions by the 10 most abundant species: (**a**) total and (**b**) per tree average.

**Figure 7 ijerph-20-00880-f007:**
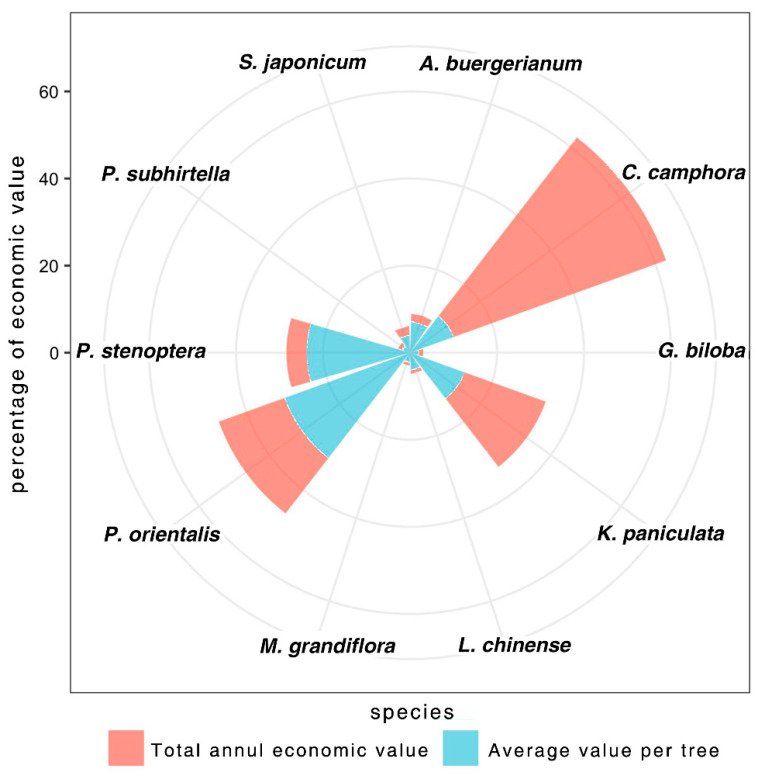
Economic value of dominant trees.

**Table 1 ijerph-20-00880-t001:** Tree species, number of trees, canopy cover, leaf area, and significant value for the 10 most abundant street trees on the JSU campus.

Species	Total Tree Numbers	Percent Total Tree Numbers	Canopy Cover (m²)	Percent Total Canopy Cover	Leaf Area (m²)	Percent Total Leaf Area	Importance Value
*Cinnamomum camphora*	1108	43.1	63,047.0	46.6	349,309.0	40.2	43.3
*Koelreuteria paniculata*	339	13.2	29,060.5	21.5	183,298.5	21.1	18.6
*Platanus orientalis*	120	4.7	10,899.8	8.1	150,878.6	17.4	10.1
*Pterocarya stenoptera*	45	1.8	6458.2	4.8	43,790.3	5.0	3.9
*Pterocarya stenoptera*	117	4.6	5371.1	4.0	18,271.3	2.1	3.6
*Ginkgo biloba*	120	4.7	2202.9	1.6	18,154.1	2.1	2.8
*Magnolia grandiflora*	126	4.9	2078.9	1.5	6783.2	0.8	2.4
*Acer buergerianum*	54	2.1	2282.5	1.7	20,140.3	2.3	2.0
*Liriodendron chinense*	50	1.9	1202.1	0.9	11,945.5	1.4	1.4
*Magnolia grandiflora*	79	3.1	824.0	0.6	4514.8	0.5	1.4
Total	2569		135,256.6		868,016.2	100.0	100.0

**Table 2 ijerph-20-00880-t002:** Carbon storage and sequestration of dominant street trees on campus at JSU.

	Carbon Storage (kg)				Carbon Sequestered (kg/Y)				Total Value (USD)
Species	Avg.	Total	Percent Total	Value (USD)	Avg.	Total	Percent Total	Value (USD)	
*C. camphora*	295.9	327,900	49.9	61,042.10	30.6	33,860	56.9	42,858.60	103,900.70
*K. paniculata*	363.7	123,300	18.7	22,953.60	21.3	7220	12.1	9138.80	32,092.40
*P. orientalis*	841.7	101,000	15.4	18,802.20	38.8	4650	7.8	5885.80	24,688.00
*P. stenoptera*	646.7	29,100	4.4	5187.70	36.4	1640	2.8	2075.80	7263.50
*S. japonicum*	101.7	11,900	1.8	2215.30	17.4	2040	3.4	2582.10	4797.50
*G. biloba*	45.3	5500	0.8	1023.90	3.4	410	0.7	519.00	1542.80
*P. subhirtella*	43.7	5500	0.8	1023.90	14.9	1880	3.2	2379.60	3403.50
*A. buergerianum*	170.4	9200	1.4	1712.70	25.6	1380	2.3	1746.70	3459.40
*L. chinense*	86.0	4300	0.6	800.50	19.4	970	1.6	1227.80	2028.30
*M. grandiflora*	57.0	4500	0.7	837.70	11.3	890	1.5	1126.50	1964.20
others		35,500	5.4	6838.40		4550	7.7	5759.30	12,597.70
Total		657,700	100	122,438.00		59,490	100	75,300.00	197,738.00

**Table 3 ijerph-20-00880-t003:** Air pollutant removal for the 10 most abundant street trees on campus at JSU.

Species	Pollutant Removal (kg/y)	Pollutant Removal (USD/y)
*C. camphora*	850.0	7807.20
*K. paniculata*	450.0	4096.80
*P. orientalis*	370.0	3372.20
*P. stenoptera*	110.0	978.70
*S. japonicum*	40.0	408.40
*G. biloba*	40.0	405.80
*P. subhirtella*	20.0	151.60
*A. buergerianum*	50.0	450.20
*L. chinense*	30.0	267.00
*M. grandiflora*	10.0	100.90
Total	2120.0	18,038.80

**Table 4 ijerph-20-00880-t004:** Runoff reduction by the most abundant street trees at JSU.

Species Name	Avoided Runoff (m³/y)	Avoided Runoff Value (USD/y)	Avg. m³ /Tree	Avg. USD/Tree
*C. camphora*	946.78	306.70	0.85	0.28
*K. paniculata*	496.82	1178.90	1.47	3.48
*P. orientalis*	408.95	970.40	3.41	8.09
*P. stenoptera*	118.69	281.60	2.64	6.26
*S. japonicum*	49.52	117.50	0.42	1.00
*G. biloba*	49.21	116.80	0.41	0.97
*P. subhirtella*	18.39	43.60	0.15	0.35
*A. buergerianum*	54.59	129.50	1.01	2.40
*L. chinense*	32.38	76.80	0.65	1.54
*M. grandiflora*	12.24	29.00	0.15	0.37
Total	2352.70	5582.60	0.92	2.17

**Table 5 ijerph-20-00880-t005:** Total economic value of dominant street trees at JSU.

Benefits	Total Y (USD)	Y (USD)/Tree
Carbon storage	128,112.50	49.90
Gross carbon sequestration	11,243.10	4.40
Pollution removal	19,477.00	7.60
Avoiding runoff	5582.60	2.17
Total value	164,415.20	64.10

## Data Availability

Not applicable.

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
