# Peer review of "Are Ecosystem Services Provided by Street Trees at Parcel Level Worthy of Attention? A Case Study of a Campus in Zhenjiang, China"

_ijerph, 2023, doi:10.3390/ijerph20010880_

Round 1
Reviewer 1 Report
Certainly the paper contains contemporary thoughts. The paper is written in clear flow and with minor syntax errors. Literature review is appropriate, but no robust explanation is given for chosen methodology.
Author Response
Please see the attachment, we have made another change.

Reviewer 2 Report
Congratulations to the authors for their work.
The work presented is novel and deals with a tremendously interesting and problematic topic in China and other countries.
In my opinion, there are a few things that need to be improved:
1. The title seems to be too general, and it is recommended that the author try to more accurately present the correspondence between the research topic and the research scope. This is a local study and should be reflected in the title.
2. The abstract is more a statement of objectives than a summary of all the parts of the article. The methodology is not explained and the conclusions are not detected. It does not cover all parts. A better structure is recommended.
3. There is some confusion between the methods and the results. In the methods, it should be clear what methodological procedures were used to arrive at the results.
4. There is an excess of references and many of them are old. I recommend reducing the number of references and leaving the most up-to-date ones. The vast majority of references should be in the introduction and although there may be some in the methodology section, there should not be so many in the final sections.
Reviewer 3 Report
The manuscript of Shen et al., is a good paper as it deals with an specific aspect of mitigation of climate change in an era of urbanization, throughout the services provided by trees. It is a novelty work and the work shows how it is important to pay attention to street trees during urban planification.
I have some suggestion authors need to take into account to improve the manuscript:
1/ Title of the paper:
I suggest: " Are ecosystem services provided by street trees at parcel-level worthy of attention?
2/ Introduction: Line 65-66 : " In addition...polluants". A reference is needed to justify the release of VOC by trees
3/ Introduction: Line 73-86: please rewrite this part by synthetizing and be concise
4/Material & Methods: Line 101-102: the link for itree tools does not work as it is wrongly reported in the manuscript
5/ Material & Methods: Line 128-129: Authors argues to conduct study between April and June 2022; but why they used pollution data from 2019 (Line146-147)?
6/ Material & Methods: Line 164-167: I suggest they displace this paragrah in introduction section.
7/ Line 180-185: Author need to indicate how they evaluate the price/ton of pollution removal value. nEED CLEARLY A REFERENCE
8/ Authors need to go in Result section Figure 3 and Table 1 indicate the same, it is a repetion
9/ Results Line 231 , figure 2 was lately indicated in the text, it is too far from the figure
